# Nurse-sensitive outcomes in district nursing care: A Delphi study

**Jessica D. Veldhuizen** [1] *, **Anne O. E. van den Bulck** [2], **Arianne M. J. Elissen** [2], **Misja C. Mikkers** [3,4], **Marieke J. Schuurmans** [5], **Nienke Bleijenberg** [1]

**1** Research Centre for Healthy and Sustainable Living, Faculty of Health Care, University of Applied Sciences Utrecht, Utrecht, The Netherlands, **2** Department of Health Services Research, Faculty of Health, Medicine and Life Sciences, Care and Public Health Research Institute (CAPHRI), Maastricht University, Maastricht, The Netherlands, **3** Dutch Healthcare Authority (NZa), Utrecht, The Netherlands, **4** Department of Economics, Tilburg School of Economics and Management, Tilburg, The Netherlands, **5** Education Center, UMC Utrecht Academy, University Medical Center Utrecht, Utrecht, The Netherlands

* Jessica.veldhuizen@HU.nl

## Abstract

### Objectives

To determine nurse-sensitive outcomes in district nursing care for community-living older people. Nurse-sensitive outcomes are defined as patient outcomes that are relevant based on nurses' scope and domain of practice and that are influenced by nursing inputs and interventions.

### Design

A Delphi study following the RAND/UCLA Appropriateness Method with two rounds of data collection.

### Setting

District nursing care in the community care setting in the Netherlands.

### Participants

Experts with current or recent clinical experience as district nurses as well as expertise in research, teaching, practice, or policy in the area of district nursing.

### Main outcome measures

Experts assessed potential nurse-sensitive outcomes for their sensitivity to nursing care by scoring the relevance of each outcome and the ability of the outcome to be influenced by nursing care (influenceability). The relevance and influenceability of each outcome were scored on a nine-point Likert scale. A group median of 7 to 9 indicated that the outcome was assessed as relevant and/or influenceable. To measure agreement among experts, the disagreement index was used, with a score of <1 indicating agreement.

**Data Availability Statement:** Data regarding the characteristics of the participants/experts cannot be shared publicly because they may be traceable, because the group is small and there are not many

experts regarding this subject. Data are available from the University of Applied Sciences Institutional Data Access (contact via onderzoekssupport@hu.nl) for researchers who meet the criteria for access to confidential data. All other data underlying the results presented in the study are available from a public repository at the Open Science Framework (OSF) via the following URL: https://osf.io/pws8r/.

**Funding:** This study was funded by the University of Applied Sciences Utrecht. The funders had no role in study design, data collection and analysis, decision to publish, or preparation of the manuscript.

**Competing interests:** The authors have declared that no competing interests exist.

## Results

In Delphi round two, 11 experts assessed 46 outcomes. In total, 26 outcomes (56.5%) were assessed as nurse-sensitive. The nurse-sensitive outcomes with the highest median scores for both relevance and influenceability were the patient's autonomy, the patient's ability to make decisions regarding the provision of care, the patient's satisfaction with delivered district nursing care, the quality of dying and death, and the compliance of the patient with needed care.

## Conclusions

This study determined 26 nurse-sensitive outcomes for district nursing care for community-living older people based on the collective opinion of experts in district nursing care. This insight could guide the development of quality indicators for district nursing care. Further research is needed to operationalise the outcomes and to determine which outcomes are relevant for specific subgroups.

## Introduction

Worldwide, healthcare services are challenged by the rapidly growing ageing population [1]. Moreover, the majority of older people desire to continue living at home, resulting in a rise in the total number of community-living older people. In Europe, the majority of older people live independently at home, either alone or with a spouse or other family members [2]. However, with increasing age, adverse consequences such as frailty, disability, chronic diseases, and multiple complex long-term conditions are present among these community-living older people [3, 4]. Because of these adverse consequences, community-living older people often need assistance with their daily life activities to be able to live at home as long as possible. Professional care assistance at home is provided through district nursing care, next to other healthcare professionals such as the general practitioner and other (paramedic) professionals in primary care [5]. The funding, organisation, definition, and delivery of district nursing care vary between countries worldwide [6–8]. For the purpose of this paper, district nursing care is defined as any technical, medical, supportive or rehabilitative nursing care and the provision of assistance with personal care [7]. This definition is in line with the definition used for community care nursing in Europe [7, 9] and reflects district nursing care in the Netherlands [10].

In many European countries, the quality of care at home is under pressure, as demands on district nursing care are increasing due to the ageing population, the increase in care complexity, and the shortage of district nursing care professionals [11, 12]. Therefore, it is crucial to monitor the quality of district nursing care in terms of patient outcomes. Insight into patient outcomes is necessary to measure the effect of healthcare services on patient health and wellbeing [13, 14]. However, patient outcomes to measure the quality of district nursing care in clinical practice on patients' health status and wellbeing are currently scarce [15].

For district nursing care, it is necessary to determine nurse-sensitive outcomes, i.e., patient outcomes that are *relevant* based on nurses' scope and domain of practice and that are *influenced* by nursing inputs and interventions [16]. The Nursing Outcome Classification (NOC) provides a set of nursing outcomes that can be used across the care continuum to assess the outcomes of care following nursing interventions [17]. However, in this overview, it is unclear what outcomes are relevant for district nursing care. Two studies, one by the International

Consortium for Health Outcomes Measurement (ICHOM) [18] and the other by Joling et al. [15] have already been conducted on outcomes that are potentially relevant to district nursing care. The ICHOM developed a set of standard health outcome measures to guide the improvement of the quality of care for the general population of older people [18]. While this study provided a meaningful overview of relevant outcomes for this population, it remains unclear whether these outcomes are nurse-sensitive outcomes specifically for district nursing care because they were developed by teams of physician leaders, researchers and patient advocates [18]. The systematic review by Joling et al. [15] identified 567 quality indicators for older people in the community care setting (i.e., primary care and district nursing care). Most of these indicators refer to care processes (80%), while only 33 indicators focus on 18 unique patient outcomes regarding health status and wellbeing (5.8%) [15]. However, it is unclear which of the proposed outcomes in the literature could be used as nurse-sensitive outcomes for district nursing care. Before quality indicators can be developed and operationalized, it is necessary to determine what outcomes are relevant to measure.

The aim of this study was to determine nurse-sensitive outcomes for district nursing care for community-living older people. Measuring nurse-sensitive outcomes for district nursing care is important because it can contribute to understanding the internal quality of teams and organisations. It provides insight into the quality of delivered care, which consequently could guide monitoring and improve the quality of district nursing care. Moreover, public transparency regarding outcomes allows patients to compare and choose a desired organisation. Finally, insight into nurse-sensitive outcomes could guide health insurers in contracting district nursing care organisations based on the quality of delivered care.

## Materials and methods

### Design

A Delphi study following the RAND/UCLA Appropriateness Method (RAM) [19] was performed. The objective of the RAM is to detect when experts agree rather than to reach consensus among experts [19]. The RAM is focused on combining available scientific evidence with the collective judgement of experts to provide a statement regarding the appropriateness of delivered care [19]. This focus fits the aim of this study to determine nurse-sensitive outcomes for district nursing care based on the collective opinion of national experts. Because of the specific national context of district nursing care, this study focused on the situation in the Netherlands. To enhance the robustness of this study, the guidance on conducting and reporting Delphi studies (CREDES) was followed [20]. In accordance with the RAM, the following steps were conducted: questionnaire development, identification of experts, two rounds of data collection (an online questionnaire and an expert panel meeting including a paper questionnaire), and data analysis after both rounds. Attrition bias due to the exhaustion of the experts was prevented by limiting the number of Delphi rounds to two rounds.

### Questionnaire development

The questionnaire was developed by reviewing the literature. Scientific and grey literature were searched using the following keywords and their accompanying synonyms: "patient outcomes," "district nursing care," and "quality indicators." For scientific literature, MEDLINE/PubMed and CINAHL/EBSCO were searched. For grey literature, international and national websites and reports of governments and research institutions were searched. Additionally, Dutch reports on what older people find important in the care that they receive at home were identified and analysed to include the patient perspective and guide the identification of important patient outcomes for district nursing care [21, 22]. The literature was reviewed until

no new outcomes for district nursing care were identified. In total, 41 patient outcomes were identified. The 41 outcomes were clustered following the domains used in the nursing outcomes classification by Moorhead et al. [17]: Functional health (n = 4), physiologic health including neurocognitive health (n = 16), psychosocial health (n = 4), health knowledge and behaviour (n = 6), perceived health (n = 2), and family health (n = 1). Additionally, the domains death (n = 2) and healthcare utilization (n = 6) were added. These outcomes were extracted from systematic reviews; peer-reviewed scientific publications, including those from the ICHOM; and reports on potentially preventable complications (see S1 Appendix). Different references were used for defining the outcomes. The outcomes were defined based on the definition used by one references or–in case definitions were incomplete, inconsistent between references, or not suitable for district nursing practice–a combination of multiple references. Because the participants were from the Netherlands, mostly Dutch literature has been used. Because the study aims to determine what outcomes are nurse-sensitive to district nursing care rather than developing and operationalizing quality indicators, the definitions of the outcomes were not constructed as quality indicators.

To determine the sensitivity of the identified outcomes to nursing care, the relevance and influenceability of the outcomes were scored. Relevance was operationalised as "being a relevant patient outcome to measure the quality of district nursing care," and influenceability was operationalised as "the extent to which district nursing care has an influence on the patient outcome."

At the beginning of the developed questionnaire, information was provided about the study. The background information of the participants regarding their age, sex, years of experience in district nursing care, and area of work was collected. Next, all 41 potential nurse-sensitive outcomes were presented along with their definitions. Participants were asked to score both the relevance and influenceability of each outcome on a 9-point Likert scale, with 1 being completely not relevant/influenceable and 9 being completely relevant/influenceable. An example question is shown in S2 Appendix. Participants had the opportunity to propose additional outcomes in case outcomes had been omitted. The complete questionnaire is available upon request.

## Identification of experts

A purposive sample of national participants was selected for the expert panel of this Delphi study. To ensure the diversity of the district nursing care professionals, the following inclusion criteria were used: 1) the participant had current or recent clinical experience as a district nurse, and 2) the participant had experience in research, teaching, practice, or policy with regard to district nursing care. The aim was to purposively create a balance between people currently working in district nursing care and those with recent experience in practice yet currently fulfilling a role in research, teaching, practice or policy regarding district nursing care. With the requirement of the nurses to have an (additional) role in research, teaching, practice, or policy, it was assumed that the nurses would be accustomed to critical thinking and reflection, which was necessary given the challenges of defining outcomes of care [16]. Participants (hereafter referred to as experts) from a diversity of organisations across the Netherlands were selected. Based on the RAM, the aim was to include a panel of 10–15 experts, which would allow the expert panel to have sufficient diversity while also ensuring that all experts would have a chance to participate [19]. To take into account the possible decline in participation during the multiple rounds, a total of 20 experts were approached via the Dutch nurses' association and the researchers' networks. Experts were informed about the study and invited to participate by email.

## Data collection

**Delphi round one: Online questionnaire.** The first Delphi round started with an online questionnaire using the online tool Qualtrics [23]. The experts received a personal invitation to the questionnaire by email. A letter including information about the study and providing consent for the study was provided within the questionnaire. The experts were asked to complete the questionnaire within two weeks. Two reminders were sent to increase the response rate. After the deadline, the online questionnaire was closed, and the results were analysed. New outcomes proposed by the experts were reviewed by a part of the research team (JDV, NB, MJS). The team discussed if the outcomes focused on patient outcomes or were relevant for measuring the quality of care. Decisions were made based on the expertise of the research team. Five outcomes were included in the next round: a meaningful life, duration of district nursing care, the intensity of district nursing care, total time at home, and quality of dying and death. Two outcomes focusing primarily on process or structure of care (providing preventive care and accessibility of district nursing team) were not included. The newly added outcomes were defined using the literature and by insights of the experts. (S1 Appendix).

**Delphi round two: Expert panel meeting and paper questionnaire.** After the analysis of the results of round one, the content from the online questionnaire was supplemented with the five newly added outcomes in a paper questionnaire. In the second Delphi round, the experts participated together in a three-hour face-to-face expert meeting. During this meeting, the findings from the questionnaire from round one regarding the relevance and influenceability of the outcomes were discussed, with special attention to the outcomes that lacked agreement (disagreement index (DI) ≥1), the outcomes that had an uncertain rating (group median 4–6), and the newly added outcomes. Additionally, the definitions of the newly added outcomes, formulated by the research team were discussed and concluded with the experts in the second Delphi round to assure that this corresponded to what the experts initially meant. After discussion of the outcomes in the expert meeting, the paper questionnaire was completed. In this questionnaire, the experts' individual scores from the first round; the group median score; and the DI, as an indication of the level of agreement, were provided (S2 Appendix).

After the analysis of the results of round two, a draft of the results was shared with the participating experts as a member check to confirm the credibility of the results.

## Data analysis

All analyses were guided by the RAM. The relevance and influenceability of each potential nurse-sensitive outcome was scored on a nine-point Likert scale. For each outcome, a group median score was calculated to determine the degree of relevance and influenceability, and the DI was calculated to determine the level of agreement. As described in the RAM, the DI is the ratio between the interpercentile range (IPR) and the IPR adjusted for symmetry (IPRAS), which can be calculated following the equation in S3 Appendix [19]. A DI <1 indicates agreement, with a score closer to zero indicating stronger agreement. A group median score of 1–3 with agreement (DI<1) indicated that the outcome was not relevant/influenceable, a lack of agreement (DI≥1) and/or a group median score of 4–6 with agreement (DI<1) on an outcome indicated that the relevance/influenceability of the outcome was uncertain, and a group median of 7–9 with agreement (DI<1) indicated that the outcome was relevant/influenceable [19]. Scores were analysed using SPSS version 24.

## Ethical considerations

The experts were informed that participation was voluntary and that all data would be processed anonymously and only for research purposes. The experts' consent was assumed upon

their return of the completed questionnaires. Because participants in this study were not subjected to physical and/or psychological procedures, no approval was needed according to the Dutch Medical Research Act (WMO). This study was conducted in accordance with the principles of the Declaration of Helsinki, and data were handled according to the General Data Protection Regulation.

## Results

### Demographics of the expert panel

In total, 16 of the 20 contacted experts (80%) agreed to participate, 15 of whom completed the online questionnaire in round one (93.8%) (Table 1). Of the experts who completed the questionnaire in round one, 11 were able to participate in the expert meeting and questionnaire in round two (73.3%). In both rounds, seven experts indicated that they worked in multiple areas of district nursing care. Reasons for non-response were a lack of time for participation and illness.

### Delphi round one

The 41 potential nurse sensitive outcomes identified in the literature were assessed by the experts in round one. The group median scores and DIs for the relevance and influenceability of the potential nurse-sensitive outcomes are provided in Table 2. Based on the median scores and DIs <1, the experts assessed 22 outcomes as relevant (53.7%) and two outcomes as not relevant (multimorbidity and planned hospital admission) (4.9%). For the remaining 17 outcomes (41.5%), there was uncertainty; for four of these outcomes, the uncertainty was due to a lack of agreement among experts.

Regarding influenceability, the experts assessed nine outcomes as influenceable (22.0%) and two outcomes as not influenceable (multimorbidity and planned hospital admission) (4.9%). The remaining 30 outcomes were assessed as uncertain (73.2%), with none lacking expert agreement. After round one, the following five outcomes were added as new outcomes: meaningful life, duration of district nursing care, intensity of district nursing care, total time at home, and quality of dying and death.

**Table 1. Characteristics of the expert panel.**

| | Delphi round 1 N = 15 | Delphi round 2 N = 11 |
|---|---|---|
| Response rate, n (%) | 15/16 (93.8) | 11/15 (73.3) |
| Age in years, mean (minimum-maximum; sd) | 40.3 (27–65; 12.2) | 35.5 (27–53; 9.2) |
| Female, n (%) | 13 (86.7) | 9 (81.8) |
| Years of clinical experience in district nursing care, mean (minimum-maximum; sd) | 12.3 (3–20; 6.4) | 10.3 (3–20; 6.0) |
| Current area of work[A] | | |
| District nurse, n (%) | 7 (46.7) | 7 (63.6) |
| Researcher, n (%) | 5 (33.3) | 3 (27.3) |
| Teacher in a bachelor of nursing program, n (%) | 5 (33.3) | 4 (36.4) |
| Practice or policy (manager, professional association), n (%) | 7 (46.7) | 6 (54.5) |

[A] The percentages do not add to 100% because some experts worked in multiple area

**Table 2. Median scores and DIs of the relevance and influenceability of outcomes per Delphi round.**

| | Relevant | | Influenceable | |
|---|---|---|---|---|
| | Round 1 Group median (DI)[A] | Round 2 Group median (DI)[A] | Round 1 Group median (DI)[A] | Round 2 Group median (DI)[A] |
| **Functional health** | | | | |
| Activities of daily living | 8 (0) | 8 (0) | 6 (0.21) | 7 (0) |
| Frailty | 7 (0) | 7 (0.22) | 6 (0.22) | 7 (0) |
| Instrumental activities of daily living | 7 (0.13)[D] | 7 (0.16) | 6 (0.72) | 6 (0.21) |
| Mobility | 7 (0.32) | 7 (0.16) | 6 (0.21) | 7 (0) |
| **Physiologic health including neurocognitive health** | | | | |
| Bladder continence | 6 (1.36)[B] | 4 (0.97) | 4 (0.32) | 4 (0.32) |
| Bowel continence | 5 (0.93) | 4 (0.52) | 4 (0.32) | 4 (0.32) |
| Cognitive functioning | 6 (0.95) | 4 (0.97) | 5 (0.32) | 5 (0.32) |
| Communication | 6 (0.86) | 4 (0.21) | 5 (0.72) | 6 (0.85) |
| Decision making | 8 (0.13) | 8 (0) | 7 (0.16) | 8 (0.16) |
| Decubitus | 8 (0.16) | 8 (0) | 7 (0.16) | 7 (0.16) |
| Dehydration | 8 (0.33) | 8 (0) | 7 (0.22) | 7 (0) |
| Delirium | 6 (0.86) | 7 (0.16) | 5 (0.97) | 7 (0.21) |
| Dyspnoea | 6 (0.95) | 6 (0.52) | 5 (0.85) | 6 (0) |
| Fatigue | 6 (0.18) | 7 (0.16) | 6 (0.32) | 7 (0) |
| Fracture and wounds other than decubitus | 6 (0.52) | 7 (0.22) | 6 (0.25) | 6 (0) |
| Infection | 7 (0.22) | 7 (0) | 6 (0) | 6 (0) |
| Multimorbidity | 3 (0.33) | 2 (0.16) | 2 (0.16) | 2 (0.16) |
| Pain | 7 (0.16) | 7 (0.16) | 7 (0.22) | 7 (0) |
| Polypharmacy | 5 (1.70)[B] | 3 (0.37) | 4 (0.98) | 4 (0.32) |
| Unintentional weight loss | 7 (0.33) | 8 (0.16) | 6 (0.45)[D] | 7 (0.37) |
| **Psychosocial health** | | | | |
| Anxiety | 6 (0.52) | 7 (0.32) | 5 (0.52) | 7 (0.22) |
| Loneliness | 7 (0.22) | 7 (0) | 5 (0.86) | 6 (0.22) |
| Participation in social activities | 7 (0.22)[D] | 7 (0) | 6 (0.18) | 7 (0.22) |
| Signs of depression | 6 (0.52) | 6 (0.51) | 5 (0.72) | 6 (0.22) |
| **Health knowledge and behaviour** | | | | |
| Autonomy | 8 (0) | 8 (0) | 7 (0.13) | 8 (0.16) |
| Compliance | 8 (0.16) | 8 (0.16) | 7 (0.13) | 8 (0.16) |
| Falls | 7 (0.32) | 8 (0.16) | 6 (0.52) | 7 (0.21) |
| Knowledge of the patient | 6 (0.49) | 2 (0.16) | 5 (0.72) | 4 (0.52) |
| Problem behaviour | 5 (0.85) | 4 (0.21) | 5 (0.72) | 5 (0.32) |
| Substance use | 4 (0.97)[D] | 3 (0.16) | 4 (0.32) | 4 (0) |
| **Perceived health** | | | | |
| Quality of life | 8 (0.16) | 8 (0.16) | 6 (0.22)[D] | 7 (0) |
| Satisfaction with district nursing care | 8 (0.23) | 8 (0) | 8 (0.16) | 8 (0.16) |
| Meaningful life[C] | - | 8 (0) | - | 7 (0.16) |
| **Family health** | | | | |
| Informal caregiver burden | 8 (0) | 8 (0) | 7 (0.16) | 7 (0) |
| **Death** | | | | |
| Death | 5 (1.36)[B] | 3 (0.16) | 4 (0.86) | 3 (0) |
| Place of death | 8 (0.16) | 8 (0.16) | 7 (0) | 7 (0.16) |
| Quality of dying and death[C] | - | 8 (0) | - | 8 (0.16) |
| **Healthcare consumption** | | | | |

*(Continued)*

**Table 2.** (Continued)

| | Relevant | | Influenceable | |
|---|---|---|---|---|
| | **Round 1 Group median (DI)[A]** | **Round 2 Group median (DI)[A]** | **Round 1 Group median (DI)[A]** | **Round 2 Group median (DI)[A]** |
| Emergency department or service use | 7 (0.37) | 7 (0) | 6 (0.42) | 7 (0) |
| General practitioner visit | 5 (0.85) | 5 (0.52) | 6 (0.72) | 6 (0.52) |
| Nursing home admission | 6 (2.38)[B] | 5 (0.96) | 6 (0.93) | 7 (0) |
| Planned hospital admission | 2 (0.37) | 2 (0) | 3 (0.59)[D] | 3 (0) |
| Unplanned hospital admission | 8 (0.65) | 8 (0.16) | 6 (0.32) | 7 (0) |
| Unplanned hospital readmission | 8 (0.33) | 8 (0) | 6 (0.22) | 7 (0.22) |
| Duration of district nursing care[C] | - | 7 (0.22) | - | 7 (0.16) |
| Intensity of district nursing care[C] | - | 7 (0.22) | - | 8 (0.16) |
| Total time at home[C] | - | 5 (0.96) | - | 6 (0.22) |

Notes

ADL: activities of daily living; IADL: instrumental activities of daily living

Green: Indicates the outcome is relevant/influenceable based on a median score between 7–9 and a DI <1.

Yellow: Indicates the uncertainty of the relevance/influenceability of the outcome based on a median score between 4–6 and/or a DI ≥1.

Red: Indicates the outcome is not relevant/influenceable based on a median score between 1–3 and a DI <1.

[A] DI: disagreement index, with a DI <1 indicating agreement.

[B] No agreement based on a DI ≥1.

[C] Newly added outcomes after Delphi round one.

[D] In an additional analysis, the median scores and DIs of round 1 with all experts (N = 15) were compared to those of round 1 with only the experts who participated in the expert meeting (N = 11). This comparison revealed the following deviating results for N = 11 compared to N = 15, as described in this table:

- IADL: DI 1.61 (uncertain relevance)
- Substance use: median 3 (not relevant)
- Participation in social activities: median 6 (uncertain relevance)
- Unintentional weight loss: median 7 (influenceable)
- Quality of life: median 7 (influenceable)
- Planned hospital admission: median 4 (uncertain influenceability)

## Delphi round two

After the face-to-face discussion in round two, the experts assessed 30 of 46 outcomes as relevant (65.2%), which were mainly distributed among the domains of functional health (4/4), perceived health (3/3), family health (1/1), psychosocial health (3/4), and outcomes regarding death (2/3). (Table 2). Six outcomes were assessed as not relevant (13.0%). The remaining 10 outcomes were assessed as uncertain (21.7%), of which none lacked expert agreement. The discussion during the expert meeting led to changes in the assessment of the relevance of eight outcomes.

Regarding influenceability after Delphi round two (Table 2), the experts assessed 27 outcomes as influenceable (58.7%), which were mainly distributed among the domains of perceived health (3/3), family health (1/1), functional health (3/4), healthcare consumption (6/9), and outcomes regarding death (2/3). Three outcomes were assessed as not influenceable (6.5%), and 16 outcomes were assessed as uncertain (34.8%). The expert meeting discussion led to changes in the assessment of the influenceability of 15 outcomes.

To determine whether the different compositions of the experts in the two rounds resulted in deviating overall results regarding the relevance and influenceability of the variables, the median scores and DIs of round 1 with all experts (N = 15) were compared to those of round 1 with only the experts who participated in the expert meeting (N = 11). This comparison revealed deviating results for the following six variables: the relevance of instrumental activities

**Table 3. Nurse-sensitive outcomes according to district nursing care experts.**

| Outcomes with a group median score of 8 for both relevance and influenceability (N = 5) | Outcomes with a group median score of 8 for relevance and 7 for influenceability (N = 12) | Outcomes with a group median score of 7 for both relevance and influenceability (N = 9) |
|---|---|---|
| • Autonomy<br>• Decision making<br>• Satisfaction with district nursing care<br>• Quality of dying and death<br>• Compliance | • ADL<br>• Dehydration<br>• Informal caregiver burden<br>• Decubitus<br>• Meaningful life<br>• Quality of life<br>• Unplanned hospital readmission<br>• Falls<br>• Unplanned hospital admission<br>• Place of death<br>• Unintentional weight loss<br>• Intensity of district nursing care[A] | • Emergency department or service use<br>• Pain<br>• Mobility<br>• Fatigue<br>• Participation in social activities<br>• Frailty<br>• Delirium<br>• Anxiety<br>• Duration of district nursing care |

ADL: activities of daily living

[A] Median score of 7 for relevance and 8 for influenceability

of daily living (IADL), substance use, and participation in social activities and the influence-ability of unintentional weight loss, quality of life and planned hospital admission. The relevance of IADL and participation in social activities changed from relevant to uncertain, and that of substance use changed from uncertain to not relevant; the influenceability of unintentional weight loss and quality of life changed from uncertain to influenceable, and that of planned hospital admission changed from not influenceable to uncertain. All other variables (92.6%) had minor changes that did not influence the overall results.

In total, the experts agreed that 26 outcomes (56.5%) were nurse-sensitive, i.e., both relevant and influenceable. From high to low, the nurse-sensitive outcomes were distributed among the following domains: perceived health (3/3), family health (1/1), functional health (3/4), death (2/3), healthcare utilization (5/9), health knowledge and behavior (3/6) psychosocial health (2/4), and physiologic health (7/16). Table 3 shows an overview of the nurse-sensitive outcomes, listed in order of most relevant and influenceable (left column) to least relevant and influence-able (right column) based on the group median and the overall DI. The nurse-sensitive outcomes with the highest median scores were the autonomy of the patient, the patient's ability to make decisions regarding the provision of care, the patient's satisfaction with delivered district nursing care, the quality of dying and death, and the compliance of the patient with needed care (i.e., the extent to which the behaviour of a patient matches the established care).

## Discussion

This study is the first to provide insight into nurse-sensitive outcomes for district nursing care based on the collective opinion of experts who represent the district nursing profession. After two Delphi rounds, the experts determined that 26 of 46 outcomes (56.5%) were nurse-sensitive outcomes for district nursing care. The nurse-sensitive outcomes that were assessed as the most relevant and influenceable (i.e., with a median of 8 and a DI between 0 and 0.16) were patient autonomy, the ability of the patient to make decisions regarding the provision of care, the patient's satisfaction with delivered district nursing care, the quality of dying and death, and the compliance of the patient with needed care.

In the comparison of our results to the outcomes of care for district nursing care described by previous studies by Joling et al. [15] and the ICHOM [18], similarities were found in 14 of

the 26 nurse-sensitive outcomes. Activities of daily living, falls, pain, participation in social activities, and informal caregiver burden were considered important outcomes by all three studies. Additionally, overlap with Joling et al. [15] was found for outcomes including decubitus, unintentional weight loss, emergency department or service use, and unplanned hospital (re)admissions. Additionally, overlap was found with the ICHOM study in relation to outcomes including autonomy, frailty, decision making, and place of death [18]. An important difference was that the experts agreed that polypharmacy and mortality were not suitable as nurse-sensitive outcomes for district nursing care. A possible explanation for the differences between our study and those by Joling et al. [15] and the ICHOM [18] lies in the focus of this Delphi study on nurse-sensitive outcomes. The other two studies did not study the relevance of these outcomes to measure the quality of district nursing care specifically and the influence nurses could or could not have on these patient outcomes. Additionally, our Delphi study determined 12 additional nurse-sensitive outcomes that were considered important and that were added by the experts after round one or were mentioned in other relevant literature on patient-reported outcomes for adults in general [24], home care quality indicators [25], or effect measures for primary care [26]. All outcomes identified in our study as nurse-sensitive outcomes for district nursing care are available as nurse outcomes in the nursing outcome classification, except for the outcomes regarding healthcare utilization, which are not included in this classification [17]. In our study, healthcare utilization was used as an outcome following other literature [15, 18].

## Strengths and limitations

To enhance the robustness of this study, the RAM and the guidance on CREDES were followed [19, 20]. An important strength was the high response rates for both rounds (93.8% and 73.3%). The differences in characteristics between the experts in the two rounds were minimal, and additional analyses showed that these differences did not influence the results for 92.6% of the variables. Additionally, the member check did not result in any comments. Furthermore, through the inclusion of experts who had clinical experience as district nurses and who had fulfilled additional roles in research, teaching, practice, or policy, the full scope of the district nursing care profession were reflected. In the interpretation of the results, some limitations should be considered. First, only Dutch experts were included in this study because of the specific district nursing context in the Netherlands. This approach limits the generalisability of the results. Second, patients were not included as experts because of the challenges regarding defining outcomes of care [16]. To incorporate their meaningful views, however, we included Dutch reports on what patients find important in receiving care at home [21, 22]. Last, the identification and definitions of the outcomes have some limitations. It is possible that outcomes and quality indicators were missed since no systematic review has been conducted. This risk was minimised by letting experts add and define missing outcomes. However, the definitions by the experts may not be comprehensive and requires further research. Additionally, the outcomes used in this study focus on older people which may limit application in district nursing care which also include care for children and middle-aged people. However, 75% of the people receiving district nursing care in The Netherlands is 67 years or older, and the mean age of the people receiving district nursing care is 75 years [27].

## Conclusion and implications

This study provides insight into nurse-sensitive outcomes based on the collective opinion of experts who represent the district nursing profession. In total, 26 nurse-sensitive outcomes were identified that could guide the development of quality indicators for district nursing care.

Measuring nurse-sensitive outcomes provides insight into the impact of district nursing care, which is a first step in monitoring and improving the quality of care. This contributes to the major call to action internationally on prioritizing the development of the evidence base for district nursing care [6]. At the national level, policy makers, the Dutch Nurses Association and healthcare organizations are working together to define quality indicators for district nursing care. The results of this study contribute to this development by determining 26 nurse-sensitive outcomes. To use nurse-sensitive outcomes as quality indicators, outcomes should be made measurable in a way that is feasible for current practice. Although the outcomes were defined based on the literature, they were not operationalized as quality indicators with a denominator and numerator. Making these nurse-sensitive outcomes measurable as quality indicators requires further research and development before their implementation in practice. In addition, the nurse-sensitive outcomes may differ between different groups of patients in various types of district nursing care, such as palliative care, rehabilitative care, and chronic care. The distinction between these groups and the accompanying relevant and influenceable outcomes for the quality of district nursing care require further research. Lastly, careful consideration is needed regarding the influenceability of the outcomes. None of these outcomes was assessed as completely relevant or influenceable (median 9), the uncertainty of the influenceability of the outcomes is relatively high (34,8%) and the overall medians of the influenceability of the outcomes are lower compared to the assessment of the relevance. This could be explained by the multidisciplinary role of district nurses in practice. Care for community-living older people is not only provided by district nurses, but also by the general practitioner and other (paramedic) professionals in primary care. Most of the outcomes are indeed often not completely influenceable by the delivered district nursing care. Coordinated care by interdisciplinary teams is associated with better outcomes regarding hospitalizations, emergency department visits, and long-term care admissions in community-living people [5]. Therefore, close collaboration between professionals in district nursing practice is needed to influence and achieve the best possible outcomes for people receiving district nursing care.

## Supporting information

**S1 Appendix. Overview of identified potential nurse-sensitive outcomes, corresponding definitions and references.**
(DOCX)

**S2 Appendix. Examples of questionnaire questions round one and round two.**
(DOCX)

**S3 Appendix. Equation to calculate disagreement index (DI).**
(DOCX)

**S4 Appendix. CREDES checklist.**
(DOCX)

## Acknowledgments

The authors thank the experts participating in this study for their time and valuable input during the survey and expert meeting. The authors also thank the academic partners from Maastricht University and the Dutch Healthcare Authority for their valuable input on this work during meetings of the Dutch Healthcare Authority's Scientific Programme on District Nursing Care.

## Author Contributions

**Conceptualization:** Jessica D. Veldhuizen, Marieke J. Schuurmans, Nienke Bleijenberg.

**Data curation:** Jessica D. Veldhuizen.

**Formal analysis:** Jessica D. Veldhuizen, Misja C. Mikkers.

**Investigation:** Jessica D. Veldhuizen, Anne O. E. van den Bulck, Nienke Bleijenberg.

**Methodology:** Jessica D. Veldhuizen, Anne O. E. van den Bulck, Arianne M. J. Elissen, Misja C. Mikkers, Marieke J. Schuurmans, Nienke Bleijenberg.

**Project administration:** Jessica D. Veldhuizen.

**Supervision:** Misja C. Mikkers, Marieke J. Schuurmans, Nienke Bleijenberg.

**Visualization:** Jessica D. Veldhuizen, Anne O. E. van den Bulck, Nienke Bleijenberg.

**Writing – original draft:** Jessica D. Veldhuizen.

**Writing – review & editing:** Jessica D. Veldhuizen, Anne O. E. van den Bulck, Arianne M. J. Elissen, Misja C. Mikkers, Marieke J. Schuurmans, Nienke Bleijenberg.

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
