## [Decision Letter · Decision Letter 0]

8 Dec 2020

PONE-D-20-16928

Nurse-sensitive outcomes in district nursing care: a Delphi study

PLOS ONE

Dear Dr. Veldhuizen,

Thank you for submitting your manuscript to PLOS ONE. After careful consideration, we feel that it has merit but does not fully meet PLOS ONE’s publication criteria as it currently stands. Therefore, we invite you to submit a revised version of the manuscript that addresses the points raised during the review process.

In particular, this paper would benefit from additional rigour in reporting and this should include justification for the methodological choices made within the Delphi rounds.  The reviewers also comment that this paper needs international appeal, so a clearer explanation of the role of the District Nurse in other countries or contexts would be helpful.

We look forward to receiving your revised manuscript.

Kind regards,

Fiona Cuthill, PhD

Academic Editor

PLOS ONE

Journal Requirements:

2.We note that you have indicated that data from this study are available upon request. PLOS only allows data to be available upon request if there are legal or ethical restrictions on sharing data publicly. For information on unacceptable data access restrictions, please see http://journals.plos.org/plosone/s/data-availability#loc-unacceptable-data-access-restrictions.

Reviewers' comments:

Reviewer's Responses to Questions

**Comments to the Author**

1. Is the manuscript technically sound, and do the data support the conclusions?

Reviewer #1: Partly

Reviewer #2: Yes

2. Has the statistical analysis been performed appropriately and rigorously? 

Reviewer #1: Yes

Reviewer #2: Yes

3. Have the authors made all data underlying the findings in their manuscript fully available?

Reviewer #1: No

Reviewer #2: No

4. Is the manuscript presented in an intelligible fashion and written in standard English?

Reviewer #1: Yes

Reviewer #2: Yes

5. Review Comments to the Author

Reviewer #1: Your paper is interesting and i enjoyed reading it. Thank you! I did not see the supporting information and perhaps some of my comments are covered in that. I offer you some comments and signposts to strengthen the paper. I hope you find them helpful.

For international appeal, identify different similar roles. The definition given does not resinate with the UK DN role, so articulate which country this reflects.

I don’t know the study is conducted in the Netherlands until p6, so perhaps this needs to be in the aim

In questionnaire development section, i’d like to understand how these indicators are worded. i know it will be in supplementary information, but perhaps including a box with some examples might help. I understand what you mean by nurse-sensitive outcomes, but i want to be convinced that the outcomes of the indicators are for patients/families living at home. On pg 13 in the Delphi Round table 2, the list on the left hand side (with no title) - are these the indicators? If so, how do e.g. falls or instrumental activities of daily living measure patient experience, health and well-being etc that you said you wanted to measure? on p17, i can see that the outcomes in the first column would be outcomes to strive for, but in the middle and the last column, would an outcome really be dehydration?

I don’t understand the terminology, instrumental activities of daily living.

p7L119 - what original reference are you referring to?

p7 L120 - If no definition was provided, other references were used - I’m not sure what this means. can you please clarify.

p7 121 - how did the authors decide the themes? Link back to what you think is missing in the outcomes literature

P8 L130 - 138 - i don’t think this fits here as experts are referred to but they haven’t been defined.

P8 L 131- who was the letter given to and how were they sampled? Why are they considered experts? these questions cos you haven’t identified experts yet.

P8 - were experts from a diversity of services/communities/organisations? Please identify.

Demographics of the expert panel on pg 11/12 would be better moved to p8 as it answers these questions. You have achieved a good balance of participants.

p9 L165 - how many outcomes were added after round one and how did the team make the decision to include them?

p12 L 214 - remind the reader how many outcomes were considered.

p15 - I can’t see C Newly added outcomes after Delphi round one in the table

p17 - Table header hat needs separated

Discussion - in the background section you have offered some critique on Joling’s and ICHOM use of outcomes and suggest that they are not dN sensitive and that they do not emphasise outcomes for patients’s health status/well-being. Rather than comparing your findings with these papers, emphasis where your findings do exactly that. So difference, rather than similarity. More literature could be used to unpack findings in this section. I think this would strengthen your discussion. What would also strengthen your discussion is how these findings could be useful in the global context to improve the quality of DN and community services. You have put this in the conclusion but it might be better here.

Limitations - there may be some limitations in the way the outcomes are written and how they were themed (this isn’t clear) I’d place the acknowledgement that these outcomes are for older people only and therefore some limitation in application to enhance DN service.

Conclusion - its great to hear these have the potential to be included in policy development. Well done!

Reviewer #2: Hello

Thank you for the opportunity to review your manuscript.

I have a few general comments and then some more specific additional comments.

Delphi studies generally require some justification as to how they choose the "expert panel". You have provided some appropriate discussion on this. I am interested to know however whether your panel were appropriate to answer your research questions. Less than half of participants in Round 1 were District Nurses (46.7% as per Table 1). Given the small numbers of participants in your Delphi rounds I think this requires additional discussion in your methods. How did you decide on numbers and sample size? What is your justification for the approach you have used? The rigour of your project relies on asking the questions to the "right" people"? Years of experience in district nursing and current area of waork does not make individuals experts. Some additional context is required to suppiort your choices.

You have used Medians to analyse all data. Do you have a justification for this? Why use Medians rather than Means? I would recommend a methodological reference to support this approach.

Providing the definitions in the Supplementary material was helful. Thankyou.

Presenting the example of Mobility provided clarity about how this was presented. I am not sure however, how "sensitive" mobility is to district nursing care. Our physiotherapist and occupational therapist colleagues would question whether mobility is truly sensitive to nursing care. When we rate the relevance of mobility as a nursing sensitive outcome for district care - what are we rating? Mobility is most frequently considered a characteristic of the person receiving care (and all patients are different with different mobility impairments). Is making someone more mobile a nursing outcome? It might be but I think we are oversimplifying this. I think improving mobility is an appropriate outcome for the healthcare team. What makes it nursing-sensitive?

I can see that an additional 5 outcomes were added after Round 1. What was the process used to define these additional outcomes? They are included as definitions in supplemental material 1 but I am unsure of the source of this definition.

See "Quality of death and dying". The definition here is "Discuss timely the options and take care of counselling in the palliative and terminal phase". This is not a very objective definition and may not actually be about "quality" of death and dying from a patient's perspective. Clarity is needed here.

I also have the following additional comments:

- Line 49. There is a 1 at the end of this first sentence? Is this meant to be a reference?

- Line 125. Nurse-sensitiveness is not an actual word. i would suggest you use 'sensitivity to nursing care' throughout your paper

- The CREDES checklist needs to be updated to reflect the page numbers for the required content (rather than the manuscript section)

There are a number of other manuscripts published that have used Delphi methods to develop a list of potential nursing-sensitive indicators. It may be useful to look at these to guide the reporting of your work.

Kind regards

The Reviewer

6. PLOS authors have the option to publish the peer review history of their article (what does this mean?). If published, this will include your full peer review and any attached files.

Reviewer #1: **Yes: **Caroline Dickson

Reviewer #2: No

---

## [Author Response · Author response to Decision Letter 0]

3 Feb 2021

REVIEWER(S)' COMMENTS TO AUTHOR

Reviewer: 1

Reviewer name: Caroline Dickson

Your paper is interesting and I enjoyed reading it. Thank you! I did not see the supporting information and perhaps some of my comments are covered in that. I offer you some comments and signposts to strengthen the paper. I hope you find them helpful.

RESPONSE: Thank you for reading this paper carefully and for your critical and helpful comments.

1. For international appeal, identify different similar roles. The definition given does not resonate with the UK DN role, so articulate which country this reflects.

RESPONS: Following your advice, we changed the definition of district nursing by adding the context of district nursing care in the introduction (page 4 line 58-63): “The funding, organisation, definition, and delivery of district nursing care vary between countries worldwide (2–4). For the purpose of this paper, district nursing care is defined as any technical, medical, supportive or rehabilitative nursing care and the provision of assistance with personal care (3). This definition is in line with the definition used for community care nursing in Europe (3,5) and reflects district nursing care in the Netherlands (6).” 

2. I don’t know the study is conducted in the Netherlands until p6, so perhaps this needs to be in the aim.

RESPONS: Thank you for your suggestion. Based on this, we have mentioned the Netherlands now in the abstract (page 2 line 28) and introduction (page 4 line 63). We did not add the country to the aim because the study generates insight into nurse-sensitive outcomes potentially suitable in other countries, providing district nursing care. 

3. In questionnaire development section, I’d like to understand how these indicators are worded.

a. I know it will be in supplementary information, but perhaps including a box with some examples might help. 

b. I understand what you mean by nurse-sensitive outcomes, but I want to be convinced that the outcomes of the indicators are for patients/families living at home. 

c. On pg 13 in the Delphi Round table 2, the list on the left hand side (with no title) - are these the indicators? If so, how do e.g. falls or instrumental activities of daily living measure patient experience, health and well-being etc. that you said you wanted to measure? on p17, I can see that the outcomes in the first column would be outcomes to strive for, but in the middle and the last column, would an outcome really be dehydration?

RESPONS: Thank you for your comments.

a. SI Appendix 1 provides all information about the definition of the outcomes. We decided to move the information about the domains to earlier in the document (page 7 line 126-130) to get an idea of what kind of outcomes are identified. We decided not to add an example to the article for it to be clear and short. 

b. We want to point out that this study focuses on determining outcomes rather than operationalizing quality indicators. Because this might be unclear until it is explained in the method section, we added the following to the introduction (page 6 line 88-89): “Before quality indicators can be developed and operationalized, it is necessary to determine what outcomes are relevant to measure.” With our study, we focused on which outcomes are nurse-sensitive for patients living at home. The outcomes included at the start of our Delphi round might indeed not be nurse-sensitive for this setting, as this is the questions we addressed to our experts. Since our experts are expert in district nursing care, and our question in the Delphi rounds are formulated with a focus on the nurse-sensitivity for this particular setting, we assume that our findings relate to those patients living at home. 

c. Table 2 show the outcomes that are assessed by the experts. We added a title on the left-hand side of table 2 to clarify (page 17, line 263). We first wanted to identify important nurse-sensitive outcomes for district nursing care (i.e. subjects/topics that are important to measure the quality of the delivered care) prior to developing and operationalizing quality indicators (which are used to make outcomes measurable). We also like to underline that there is a difference between the outcome that we want to measure (e.g. presence or degree of dehydration) and the goal we strive for in the care we deliver (absence of dehydration). Further research is needed to operationalise these outcomes to quality indicators, as is described in the discussion section (page 26 line 378-383). To make this more clear and transparent, we added the following sentence to the method section (page 8 line 137-140): “because the study aims to determine what outcomes are nurse-sensitive to district nursing care rather than developing and operationalizing quality indicators, the definitions of the outcomes were not constructed as quality indicators”. 

4. I don’t understand the terminology, instrumental activities of daily living. 

RESPONS: We described all definitions in SI Appendix 1 because most outcomes are common in (district) nursing care. Instrumental activities of daily living is defined as “The extent to which the patient (together with the people around the patient) is independent in carrying out instrumental activities of daily living (IADL) such as housework, shopping, preparing meals, and making telephone calls”. We hope that we provided sufficient clarification with the definition from SI Appendix 1. 

5. Discussion 

a. in the background section you have offered some critique on Joling’s and ICHOM use of outcomes and suggest that they are not DN sensitive and that they do not emphasize outcomes for patients’ health status/well-being. Rather than comparing your findings with these papers, emphasis where your findings do exactly that. So difference, rather than similarity. More literature could be used to unpack findings in this section. I think this would strengthen your discussion. 

b. What would also strengthen your discussion is how these findings could be useful in the global context to improve the quality of DN and community services. You have put this in the conclusion but it might be better here.

RESPONS: Thank you for your suggestions. 

a. Joling and ICHOM identify important quality indicators and outcomes for community-living older people as described in the introduction (page 5 line 76-88). However, Joling and ICHOM do not focus on nurse-sensitive outcomes. To emphasize our study’s focus on nurse-sensitive outcomes, we added our study’s aim at the beginning of the discussion (page 22 line 314-315). We also connected the results of our study to the nursing outcome classification (NOC) by Moorhead et al. (7) to show that all outcomes identified in our study are nurse-sensitive (page 24 line 338-342): “All outcomes identified in our study as nurse-sensitive outcomes for district nursing care are available as nurse outcomes in the nursing outcome classification, except for the outcomes regarding healthcare utilization, which are not included in this classification (7). In our study, healthcare utilization was used as an outcome following other literature (8,9)”. The NOC is developed using various and an extensive amount of literature (7). We added background information about the NOC to the introduction (page 5 line 73-76): “The Nursing Outcome Classification (NOC) provides a set of nursing outcomes that can be used across the care continuum to assess the outcomes of care following nursing interventions (7). However, in this overview, it is unclear what outcomes are relevant for district nursing care”. 

b. Because the conclusion is part of the discussion and to prevent repeating information, we decided to leave the usefulness of our finding in the global context in the conclusion. 

6. Limitations - there may be some limitations in a. the way the outcomes are written and b. how they were themed (this isn’t clear) I’d place the acknowledgement that these outcomes are for older people only and therefore some limitation in application to enhance DN service.

RESPONS: 

a. We agree that this study’s focus on older people may be a limitation of the study. Therefore we added the following sentences to the description of the study’s limitations (page 24 line 362-366): “Additionally, the outcomes used in this study focus on older people which may limit application in district nursing care which also include care for children and middle-aged people. However, 75% of the people receiving district nursing care in The Netherlands is 67 years or older, and the mean age of the people receiving district nursing care is 75 years (10)”. 

b. In the previous version of the manuscript, we divided the outcomes in themes used in ICHOM and other references. However, based on reviewer comments and new insights, we have decided to use the NOC domains (7) instead of themes because the nursing outcome classification domains fit better in our study focusing on nurse-sensitive outcomes. We have changed “themes” into “domains” throughout the manuscript and changed the order of the outcomes in table 2 and SI Appendix 1. We added the following to the method section (page 7, line 126-130): “The 41 outcomes were clustered following the domains used in the nursing outcomes classification by Moorhead et al. (7): Functional health (n=4), physiologic health including neurocognitive health (n=16), psychosocial health (n=4), health knowledge and behaviour (n=6), perceived health (n=2), and family health (n=1). Additionally, the domains death (n=2) and healthcare utilization (n=6) were added.”

7. Conclusion - it’s great to hear these have the potential to be included in policy development. Well done!

RESPONS: Thank you very much.

Minor edits 

8. p7L119 - what original reference are you referring to?

RESPONS: The original references are described in SI Appendix 1. We added the explanation provided in SI Appendix 1 to the method section as follows (page 8 line 132-140): “Different references were used for defining the outcomes. The outcomes were defined based on the definition used by one references or – in case definitions were incomplete, inconsistent between references, or not suitable for district nursing practice – a combination of multiple references. Because the participants were from the Netherlands, mostly Dutch literature has been used. Because the study aims to determine what outcomes are nurse-sensitive to district nursing care rather than developing and operationalizing quality indicators, the definitions of the outcomes were not constructed as quality indicators”. 

9. p7 L120 - If no definition was provided, other references were used - I’m not sure what this means. can you please clarify.

RESPONS: To clarify the process of defining outcomes, we changed the text in the manuscript, as mentioned in our response to reviewer 1 - comment 8. We followed the definitions provided in the original references, for example, the outcome “place of death” was defined following the description described by ICHOM. However, definitions provided were sometimes incomplete, inconsistent between reference, or not suitable for district nursing practice. Therefore, when no reference provided a (suitable) definition, we used multiple references to combine definitions of similar outcomes into one definition to define the outcome. For example, we defined falls as “The presence of fall incidents, where a fall incident is defined as an unintended change of body position that results in a fall on the ground or another lower level” combining the definitions of ICHOM and those provided by Bakker et al. (11). This has been done before by a previous study by van den Bulck et al. (12). With the author’s permission, we used some of the definitions provided by van den Bulck et al. (12). SI Appendix 1 provides an overview of all literature used for identifying and defining the outcomes. We added the sources of the outcomes to table 1 in SI Appendix 1 (page 5-12, line 104). 

10. p7 121 - how did the authors decide the themes? Link back to what you think is missing in the outcomes literature

RESPONS: We would like to refer to our response at point 6, where we clarified this point. 

11. P8 L130 - 138 - I don’t think this fits here as experts are referred to but they haven’t been defined.

RESPONS: We agree that using the term “experts” may be confusing. Therefore we changed “experts” to “participants” in page 9 line 149-170. After line 170 we decided to use “experts”, following the RAM. After introducing the participants as experts, we called them experts to adhere to the terminology of the RAND/UCLA Appropriateness Method (RAM) User’s Manual (13). We added this to the method section as follows (line 170): “Participants (hereafter referred to as experts) […]”.

12. P8 L 131- 

a. who was the letter given to and how were they sampled? 

b. Why are they considered experts? these questions cos you haven’t identified experts yet.

RESPONS: We changed the order of the method section to make this more clear. 

a. We moved information about providing consent for the study from “questionnaire development” to “data collection” (page 10 line 182-184). Sampling has been described under “identification of experts” (page 9 line 160-176). 

b. We understand that the wording of experts may be confusing before the experts are identified. To reduce confusion, we changed “experts” to “participants” before the experts are identified (page 9 line 149-170). After introducing the participants as experts, we called them experts to adhere to the terminology of the RAND/UCLA Appropriateness Method (RAM) User’s Manual (13). We added this to the method section as follows (line 170): “Participants (hereafter referred to as experts) […]”.

13. P8 – 

a. were experts from a diversity of services/communities/organisations? Please identify.

b. Demographics of the expert panel on pg 11/12 would be better moved to p8 as it answers these questions. You have achieved a good balance of participants.

RESPONS: Thank you for your compliment regarding the balance of participants. 

a. Experts were from a diversity of organisations across the Netherlands. Because we agree that this was not clear, we added the following to the method section (page 10 line 170-171): “Participants (hereafter referred to as experts) from a diversity of organisations across the Netherlands were selected”. 

b. We agree it would be helpful to provide the demographics of the experts to the method section. However, to distinguish between study methods and the study results, we decided to leave this information in the results section. 

14. p9 L165 - how many outcomes were added after round one and how did the team make the decision to include them?

RESPONS: We agree that we could elaborate more on this subject. Therefore we added the following information regarding proposed and newly added outcomes under the heading Delphi round one, page 10 line 186-193: “New outcomes proposed by the experts were reviewed by a part of the research team (JDV, NB, MJS). The team discussed if the outcomes focused on patient outcomes or were relevant for measuring the quality of care. Decisions were made based on the expertise of the research team. Five outcomes were included in the next round: a meaningful life, duration of district nursing care, the intensity of district nursing care, total time at home, and quality of dying and death. Two outcomes focusing primarily on process or structure of care (providing preventive care and accessibility of district nursing team) were not included. The newly added outcomes were defined using the literature and by insights of the experts.” In the description of round two, the following was added: “Additionally, the definitions of the newly added outcomes, formulated by the research team were discussed and concluded with the experts in the second Delphi round to assure that this corresponded to what the experts initially meant”. 

15. p12 L 214 - remind the reader how many outcomes were considered.

RESPONS: We added the total number of outcomes at the beginning of Delphi round 1 (results section) to remind the reader (page 14, line 247-248). 

16. p15 - I can’t see C Newly added outcomes after Delphi round one in the table

RESPONS: the newly added outcomes are provided in table 2 and indicated with a C behind the outcome’s names in the first column. This relates to five outcomes included in the table, i.e. a meaningful life, duration of district nursing care, the intensity of district nursing care, total time at home, and quality of dying and death. The outcomes can be found in the table underneath the domains ‘Perceived health’, ‘Death’, and ‘Healthcare consumption’. 

17. p17 - Table header hat needs separated

RESPONS: Thank you for your attention to the details. We changed the header’s format in table 3 from centre align to align left to enhance the table's readability. 

Reviewer: 2

Reviewer name: not available

Hello, 

Thank you for the opportunity to review your manuscript. I have a few general comments and then some more specific additional comments.

RESPONSE: Thank you very much for taking the time to review our paper.

1. Delphi studies generally require some justification as to how they choose the "expert panel". You have provided some appropriate discussion on this. I am interested to know however whether your panel were appropriate to answer your research questions. 

a. Less than half of participants in Round 1 were District Nurses (46.7% as per Table 1). Given the small numbers of participants in your Delphi rounds I think this requires additional discussion in your methods. How did you decide on numbers and sample size? What is your justification for the approach you have used? 

b. The rigour of your project relies on asking the questions to the "right" people"? Years of experience in district nursing and current area of work does not make individuals experts. Some additional context is required to support your choices.

RESPONS: It is a valid point to check if the sample of experts and choice for experts were appropriate to answer the research question. 

a. The sample size has been chosen based on RAM (13), this has been described in the method section (page 10 line 171-173): “Based on the RAM, the aim was to include a panel of 10-15 experts, which would allow the expert panel to have sufficient diversity while also ensuring that all experts would have a chance to participate”. With 15 experts in the first round and 11 in the second round, we complied to this. We elaborate on the choice of the experts in point 1.b below. 

b. To support our choices regarding choosing the right people, we added the following to the method section: (page 9 line 165-167): “The aim was to purposively create a balance between participants currently working in district nursing care and those with recent experience in practice yet currently fulfilling a role in research, teaching, practice or policy regarding district nursing care”. The experts were selected based on two criteria: they had to work as a district nurse or have recent experience as a district nurse. Moreover, those who were not currently working as a district nurse were selected given their work regarding district nursing in research, teaching, practice, or policy. In our opinion, 100% of the participants were appropriate to answer the research question. We agree that years of experience does not make a person an expert. Therefore, we added the experts' criteria to have an additional role in research, teaching, practice, or policy. With this, it was assumed that the nurses would be accustomed to critical thinking and reflection, which was necessary given the challenges of defining care outcomes (14). Additionally, we purposively selected participants from a diversity of organisations. This was also added to the method section as follows (page 10 line 170-171): “participants (hereafter referred to as experts) from a diversity of organisations across the Netherlands were selected”. 

2. You have used Medians to analyse all data. Do you have a justification for this? Why use Medians rather than Means? I would recommend a methodological reference to support this approach.

RESPONS: We followed the RAM instructions (13), in which they work with medians. It is also common in Delphi studies to analyse data using medians (15). Means are more sensitive to outliers with a small sample size than medians. We agree that it could be more clearly described that the RAM guided us. Therefore, we added the following sentence to the data analysis section, page 12 line 212: “All analyses were guided by the RAM.” 

3. Providing the definitions in the Supplementary material was helpful. Thank you. Presenting the example of Mobility provided clarity about how this was presented. I am not sure however, how "sensitive" mobility is to district nursing care. Our physiotherapist and occupational therapist colleagues would question whether mobility is truly sensitive to nursing care. When we rate the relevance of mobility as a nursing sensitive outcome for district care - what are we rating? Mobility is most frequently considered a characteristic of the person receiving care (and all patients are different with different mobility impairments). Is making someone more mobile a nursing outcome? It might be but I think we are oversimplifying this. I think improving mobility is an appropriate outcome for the healthcare team. What makes it nursing-sensitive?

RESPONS: This is a valid point. We agree that the identified outcomes are not only sensitive to nursing interventions but may also be relevant outcomes for other disciplines or the whole healthcare team. This reflects in the median scores: the experts agree that the outcomes are nurse-sensitive (with a median of 7 or 8 regarding the outcomes' relevance and influenceability), but none of the outcomes is assessed as completely nurse-sensitive (median 9). We believe that this is in line with district nursing practice, in which nurses fulfil a multidisciplinary role within primary care. Care for community-living older people is provided by various professionals, such as district nurses, the general practitioner, physiotherapist, occupational therapist, or other disciplines. Therefore, most of the outcomes are indeed often not completely influenceable by the delivered district nursing care only. We believe that this reflection is essential to share. Therefore we added this point to the discussion section, page 26 line 387: “Lastly, careful consideration is needed regarding the influenceability of the outcomes. None of these outcomes was assessed as completely relevant or influenceable (median 9), the uncertainty of the influenceability of the outcomes is relatively high (34,8%) and the overall medians of the influenceability of the outcomes are lower compared to the assessment of the relevance. This could be explained by the multidisciplinary role of district nurses in practice. Care for community-living older people is not only provided by district nurses, but also by the general practitioner and other (paramedic) professionals in primary care. Most of the outcomes are indeed often not completely influenceable by the delivered district nursing care. Coordinated care by interdisciplinary teams is associated with better outcomes regarding hospitalizations, emergency department visits, and long-term care admissions in community-living people (16). Therefore, close collaboration between professionals in district nursing practice is needed to influence and achieve the best possible the outcomes for clients who receive district nursing care”. Additionally, we put emphasis on other disciplines in the introduction, page 4 line 56-58: “Professional care assistance at home is provided through district nursing care, next to other healthcare professionals such as the general practitioner and other (paramedic) professionals in primary care (16)”. 

4. I can see that an additional 5 outcomes were added after Round 1. What was the process used to define these additional outcomes? They are included as definitions in supplemental material 1 but I am unsure of the source of this definition.

RESPONS: The experts proposed the new outcomes during the first online questionnaire. The expert explained the newly proposed outcomes in this questionnaire. Based on this, the research team defined the outcomes using literature. During the expert meeting, the definitions were discussed with the experts. To explain and justify the choices made, we added the following to the method section (page 10 line 186-193): “New outcomes proposed by the experts were reviewed by a part of the research team (JDV, NB, MJS). The team discussed if the outcomes focused on patient outcomes or were relevant for measuring the quality of care. Decisions were made based on the expertise of the research team. […] The newly added outcomes were defined using the literature (SI Appendix 1)”. Additionally, we added the following sentence to the description of round 2 (page 11 line 202-204): “Additionally, the definitions of the newly added outcomes, formulated by the research team were discussed and concluded with the experts in the second Delphi round to assure that this corresponded to what the experts initially meant”. 

5. See "Quality of death and dying". The definition here is "Discuss timely the options and take care of counselling in the palliative and terminal phase". This is not a very objective definition and may not actually be about "quality" of death and dying from a patient's perspective. Clarity is needed here.

RESPONS: We agree that this definition should be studied further. As explained above, the definitions were based on the literature and insights from the experts. Regarding this outcome, the experts' insights and opinions were used because there is a lack of clarity and consistency in the literature regarding this construct (17). However, we agree that the experts in our study might not be experts regarding the subject “quality of death and dying”. In line with this, we would like to emphasise that the study aims to identify important outcomes for district nursing care. The operationalisation of the outcomes is not determined in this study and requires further research. To underline this limitation, we added the following to the limitations in the discussion section (page 25 line 357-361): “Last, the identification and definitions of the outcomes have some limitations. It is possible that outcomes and quality indicators were missed since no systematic review has been conducted. This risk was minimised by letting experts add and define missing outcomes. However, the definitions by the experts may not be comprehensive and requires further research”.

Minor edits

6. Line 49. There is a 1 at the end of this first sentence? Is this meant to be a reference?

RESPONS: Thank you for your attention regarding the details. The 1 at the end of the sentence is indeed supposed to be a reference. We changed it to the correct formatting. 

7. Line 125. Nurse-sensitiveness is not an actual word. i would suggest you use 'sensitivity to nursing care' throughout your paper

RESPONS: We changed “nurse-sensitiveness” to “sensitivity to nursing care”. 

8. The CREDES checklist needs to be updated to reflect the page numbers for the required content (rather than the manuscript section)

RESPONS: Thank you for pointing this out to us. We updated the CREDES checklist to reflect the page numbers. 

9. There are a number of other manuscripts published that have used Delphi methods to develop a list of potential nursing-sensitive indicators. It may be useful to look at these to guide the reporting of your work.

RESPONS: Thank you very much for your suggestion. We looked at other manuscripts to guide the reporting of our work in combination with the guidelines for Conducting and Reporting Delphi Studies (CREDES) (1). 

References:

1. Jünger, S; Payne, SA; Brine, J; Radbruch, L; Brearley S. Guidance on Conducting and REporting DElphi Studies (CREDES) in palliative care: Recommendations based on a methodological systematic review. Palliative Medicine. 2017;31(8):684–706. 

2. Jarrin, OF; Pouladi, FA; Madigan E. International priorities for home care education, research, practice, and management: Qualitative content analysis. Nurse Education Today. 2019;73:83–7. 

3. Van Eenoo L, Declercq A, Onder G, Finne-Soveri H, Garms-Homolova V, Jonsson PV, e.a. Substantial between-country differences in organising community care for older people in Europe—a review. The European Journal of Public Health. 2016;26(2):213–9. 

4. World Health Organization. Home care across Europe: current structure and future challenges. World Health Organization. Regional Office for Europe; 2012. 

5. Tarricone R, Tsouros AD. Home care in Europe: the solid facts. WHO Regional Office Europe; 2008. 

6. Maurits EEM. Autonomy of nursing staff and the attractiveness of working in home care [PhD Thesis]. Utrecht University; 2019. 

7. Moorhead S, Johnson M, Maas ML, Swanson E. Nursing Outcomes Classification (NOC)-e-book: Measurement of health outcomes. Elsevier Health Sciences; 2018. 

8. Joling KJ, Van Eenoo L, Vetrano DL, Smaardijk VR, Declercq A, Onder G, e.a. Quality indicators for community care for older people: A systematic review. PloS one. 2018;13(1):e0190298. 

9. Akpan A, Roberts C, Bandeen-Roche K, Batty B, Bausewein C, Bell D, e.a. Standard set of health outcome measures for older persons. BMC geriatrics. 2018;18(1):36. 

10. Vektis. Factsheet Wijkverpleging [Internet]. 2020 [geciteerd 4 januari 2021]. Beschikbaar op: https://www.vektis.nl/intelligence/publicaties/factsheet-wijkverpleging

11. Bakker AJEM, Habes V, Quist G, others. Klinisch redeneren bij ouderen: functiebehoud in levensloopperspectief. Bohn Stafleu van Loghum; 2016. 

12. van den Bulck AO, Metzelthin SF, Elissen AM, Stadlander MC, Stam JE, Wallinga G, e.a. Which client characteristics predict home-care needs? Results of a survey study among Dutch home-care nurses. Health & Social Care in the Community. 2019;27(1):93–104. 

13. Fitch, K; Bernstein, SJ; Aguilar, MD; Burnand, B; LaCalle, JR; Lázaro, P; van het Loo, M; McDonnell, J; Vader, JP; Kahan J. The RAND/UCLA Appropriateness Method User’s Manual. 2001. 

14. Doran D. Nursing outcomes. Jones & Bartlett Learning; 2010. 

15. Boulkedid R, Abdoul H, Loustau M, Sibony O, Alberti C. Using and reporting the Delphi method for selecting healthcare quality indicators: a systematic review. PloS one. 2011;6(6):e20476. 

16. Stall N, Nowaczynski M, Sinha SK. Systematic review of outcomes from home-based primary care programs for homebound older adults. Journal of the American Geriatrics Society. 2014;62(12):2243–51. 

17. Hales S, Zimmermann C, Rodin G. The quality of dying and death: a systematic review of measures. Palliative Medicine. 2010;24(2):127–44.

---

## [Decision Letter · Decision Letter 1]

29 Apr 2021

Nurse-sensitive outcomes in district nursing care: a Delphi study

PONE-D-20-16928R1

Dear Dr. Veldhuizen,

We’re pleased to inform you that your manuscript has been judged scientifically suitable for publication and will be formally accepted for publication once it meets all outstanding technical requirements.

Kind regards,

Fiona Cuthill, PhD

Academic Editor

PLOS ONE

Additional Editor Comments (optional):

Reviewers' comments:

Reviewer's Responses to Questions

**Comments to the Author**

1. If the authors have adequately addressed your comments raised in a previous round of review and you feel that this manuscript is now acceptable for publication, you may indicate that here to bypass the “Comments to the Author” section, enter your conflict of interest statement in the “Confidential to Editor” section, and submit your "Accept" recommendation.

Reviewer #1: All comments have been addressed

2. Is the manuscript technically sound, and do the data support the conclusions?

Reviewer #1: Yes

3. Has the statistical analysis been performed appropriately and rigorously? 

Reviewer #1: I Don't Know

4. Have the authors made all data underlying the findings in their manuscript fully available?

Reviewer #1: Yes

5. Is the manuscript presented in an intelligible fashion and written in standard English?

Reviewer #1: Yes

6. Review Comments to the Author

Reviewer #1: Thank you very much for addressing my comments and re-submitting your manuscript.

I wish you well with your work and hope the paper achieves some impact.

7. PLOS authors have the option to publish the peer review history of their article (what does this mean?). If published, this will include your full peer review and any attached files.

Reviewer #1: **Yes: **Dr Caroline Dickson

---

## [Editor Report · Acceptance letter]

3 May 2021

PONE-D-20-16928R1 

Nurse-sensitive outcomes in district nursing care: a Delphi study 

Dear Dr. Veldhuizen:

I'm pleased to inform you that your manuscript has been deemed suitable for publication in PLOS ONE. Congratulations! Your manuscript is now with our production department. 

Kind regards, 

on behalf of

Dr. Fiona Cuthill 

Academic Editor

PLOS ONE